# Task-dependent coarticulation of movement sequences

Hari Teja Kalidindi[1,2], Frederic Crevecoeur[1,2,3]*

[1]Institute for Information and Communication Technologies, Electronics and Applied Mathematics (ICTEAM), Université Catholique de Louvain, Louvain-la-Neuve, Belgium; [2]Institute of Neuroscience (IoNS), Université Catholique de Louvain, Brussels, Belgium; [3]WEL Research Institute, Wavre, Belgium

**Abstract** Combining individual actions into sequences is a hallmark of everyday activities. Classical theories propose that the motor system forms a single specification of the sequence as a whole, leading to the coarticulation of the different elements. In contrast, recent neural recordings challenge this idea and suggest independent execution of each element specified separately. Here, we show that separate or coarticulated sequences can result from the same task-dependent controller, without implying different representations in the brain. Simulations show that planning for multiple reaches simultaneously allows separate or coarticulated sequences depending on instructions about intermediate goals. Human experiments in a two-reach sequence task validated this model. Furthermore, in co-articulated sequences, the second goal influenced long-latency stretch responses to external loads applied during the first reach, demonstrating the involvement of the sensorimotor network supporting fast feedback control. Overall, our study establishes a computational framework for sequence production that highlights the importance of feedback control in this essential motor skill.

*For correspondence:
frederic.crevecoeur@uclouvain.be

**Competing interest:** The authors declare that no competing interests exist.

## eLife assessment

This **valuable** paper presents **convincing** evidence that changing the constraint of how long to stop at an intermediate target significantly influences the degree of coarticulation of two sequential reaching movements, as well as their response to mechanical perturbations. Using an optimal-control framework, the authors offer a normative explanation of how both co-articulated and separated sequential movement can be understood as an optimal solution to the task requirements.

## Introduction

Everyday behavior involves performing sequences of actions to achieve a given goal. Sequential motor skills depend on the ability to combine individual actions with short-term goals into a longer sequence consisting of multiple goals (*Botvinick and Plaut, 2004*). However, the computations that underlie the combination of sequence elements and the neural circuits where this combination occurs remain a topic of debate (*Diedrichsen and Kornysheva, 2015*; *Hikosaka et al., 1999*; *Wong and Krakauer, 2019*).

Behavioral data from various sensorimotor repertoires indicate that the trajectory of sub-movements in a sequence is affected by multiple future goals (*Ariani and Diedrichsen, 2019*; *Diamond et al., 2017*; *Gottwald et al., 2018*; *Sheahan et al., 2016*; *Sosnik et al., 2004*; *Kashefi et al., 2024*). For example, in human speech and bird song, the pronunciation of individual elements is influenced by subsequent elements (*Fowler and Saltzman, 1993*; *Wohlgemuth et al., 2010*), a phenomenon known as coarticulation. Ramkumar and colleagues (*Ramkumar et al., 2016*) employed a computational

model, revealing that concatenating multiple sub-movements into larger units, referred to as chunks (**Verwey, 2001**; **Verwey and Abrahamse, 2012**), induces coarticulation within the velocity patterns of sub-movements. This coarticulation serves as a natural solution to enhance the efficiency of chunked movements. Collectively, behavioral findings highlight a pattern where sub-movements depend on the whole sequence, or a chunk.

It remains unclear how future goals are integrated in the sensorimotor system. For rapid execution of a sequence, one possible solution is to represent multiple goals within low-level control circuits (**Hikosaka et al., 1999**; **Kadmon Harpaz et al., 2022**), enabling the execution of several elements as a single entity, called a 'motor chunk.' Note that chunking can also occur at a higher level such as in working memory-guided sequences, which in this case may or may not involve the production of a movement (**Mizes et al., 2023**; **Miller, 1956**).

Recent neural recordings in the primary motor cortex (M1) have shown no specific influence of future goals on the population responses governing ongoing action (**Yokoi and Diedrichsen, 2019**; **Zimnik and Churchland, 2021**). Specifically, (**Zimnik and Churchland, 2021**) observed in a two-reach sequence task that, there was no coarticulation in sub-movement kinematics although

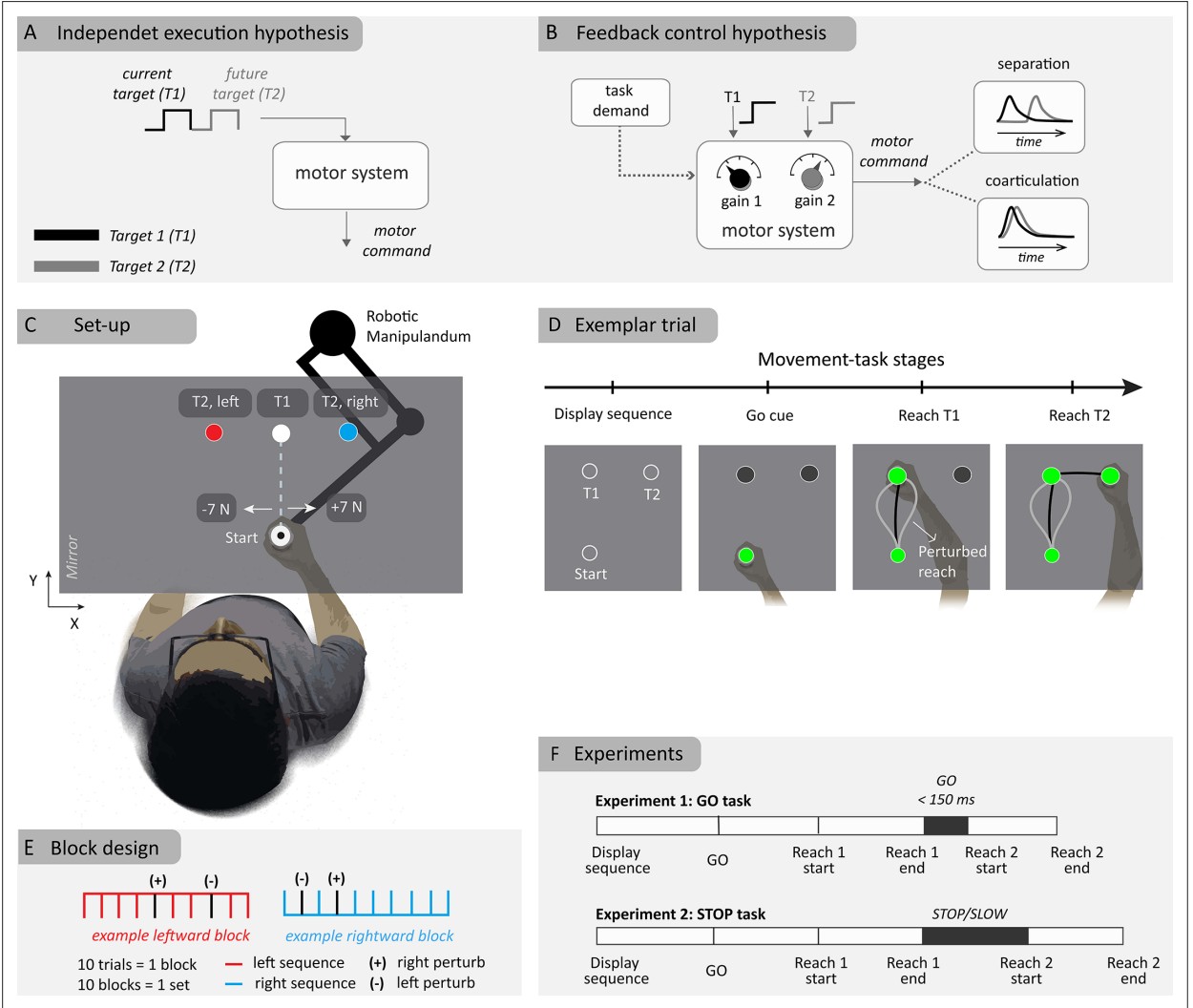

**Figure 1.** Schematic illustration of hypotheses and experiment method. (**A**) Independent control hypothesis where each target is specified one at a time onto the controller. (**B**) Flexible feedback control hypothesis where multiple targets are simultaneously considered. (**C**) Experimental setup of two-reach sequences with two different second targets (**T2**) sharing the first target (**T1**). Perturbations of 7 N magnitude (rightward or leftward) were applied close to the start location. (**D**) Sequence of events in a typical (example) trial. (**E**) Block design where the right, left-target blocks are randomly interleaved within a set where each block consist of eight unperturbed and two perturbed trials. (+) and (-) represent the direction of applied loads in perturbation trials (**F**) Experiment protocol for the 'GO' and 'STOP' tasks, with differing demands at the first target.

the execution got faster with practice. Notably, M1 displayed separate phases of execution-related activity for each sub-movement. Using a neural network model, they interpreted that sequence goals could be separated and serially specified to the controller from regions upstream of M1 (*Figure 1A*). These findings contrast with earlier studies showing coarticulation of sub-movements and whole sequence representations in M1 (*Ben-Shaul et al., 2004*; *Hatsopoulos et al., 2003*; *Lu and Ashe, 2005*). As a result, it has been suggested that coarticulation and separation in rapid sequences may involve distinct computations: coarticulation possibly involves replacing sub-movements with a motor chunk, while separation possibly indicates independent control of each sub-movement with chunking at a higher-level (*Wong and Krakauer, 2019*; *Zimnik and Churchland, 2021*). Thus, there are unresolved questions regarding why sequential movements sometimes coarticulate, and how the representation of future goals in the sensorimotor system influences the way sequences are executed.

Strikingly, most studies have emphasized potential differences in the state of the sensorimotor system during preparatory activity (*Zimnik and Churchland, 2021*; *Ben-Shaul et al., 2004*; *Hatsopoulos et al., 2003*; *Lu and Ashe, 2005*). As a consequence, the role of feedback control in this debate has been unexplored, although it is known that human motor control involves continuous processing of sensory feedback (*Crevecoeur and Kurtzer, 2018*; *Parrell and Houde, 2019*). The theory of optimal feedback control (OFC) has been particularly useful in predicting the influence of numerous task parameters on the controller (*Diedrichsen et al., 2010*; *Cluff and Scott, 2013*; *Cluff and Scott, 2015*; *Kalidindi and Crevecoeur, 2023*; *Todorov and Jordan, 2002*; *Scott, 2004*; *Kurtzer et al., 2008*; *De Comite et al., 2022*), thus reproducing goal-directed motor commands during both unperturbed movements and feedback responses to disturbances (*Kalidindi and Crevecoeur, 2023*). OFC has been used in numerous studies to interpret flexible feedback responses occurring in the long-latency response period (*Kalidindi and Crevecoeur, 2023*; *Scott, 2016*). In this framework, all targets in a sequence can be considered simultaneously by the controller, and the overlap between control gains applied to each target determines when they influence the sequence (*Figure 1B*). The control gains themselves depend on task requirements (*Liu and Todorov, 2007*; *Dimitriou et al., 2013*; *Cesonis and Franklin, 2021*). Thus, OFC predicts two experimentally testable features of sequence production: first, even if the sequence is considered as a whole at the level of controller, control gains can show different amounts of overlap based on task requirements, leading to either coarticulation or separation of the sequence elements; second, in sequences with strong co-articulation, feedback responses to perturbations are expected to depend on future targets due to the overlap of control gains.

Here, we studied these predictions of OFC in a two-reach sequence task. Simulations revealed that the instruction to slow-down at the first target influenced the coarticulation of the two reaches. Importantly, the penalty for the velocity at the first target produced an apparent separation of two reaches, even if the controller was always computed jointly for both reaches. We conducted two experiments with human participants and observed that their behavior was fully consistent with the simulations. We observed both flexible coarticulation dependent on task instruction about the intermediate target, as well as a modulation of long-latency feedback responses to the mechanical loads applied during the first reach dependent on the location of the second target when coarticulation was present. These results demonstrate that both coarticulation and separation of sub-movements may arise as the consequence of efficient feedback control. Recordings of muscle activity suggest that this operation is carried out by the sensorimotor network that mediates fast state-feedback control.

## Results

The results section is organized as follows: we first introduce the computational model and describe the results of the simulations showing that separate or coarticulated sequence production can result from flexible feedback control without implying a different structure in the controller (*Figure 1A–B*). Second, we present the two behavioral experiments in which participants were instructed to perform a sequence of two reaching movements with an intermediate goal (*Figure 1C–F*). Our key manipulation guided by the model simulations was to give them different instructions regarding the pause at the first target, which produced the modulation of sequence production predicted in theory.

## Simulations of sequence execution

The model considers a task-dependent state-feedback controller derived from the framework of stochastic optimal control (*Kalidindi and Crevecoeur, 2023*). The controller was optimized to produce a translation of a point mass on a horizontal plane. A given estimate of a state vector $\hat{x}$ (position, velocity, target location and external load) can influence the motor output $u$ through multiplicative functions called 'feedback control gains,' represented by $k(t)$. Typically, the motor output at any time ($t$) is a result of a state feedback control law:

$$u(t) = k(t)\hat{x}(t).$$

Note that the control law is dependent on the state estimate ($\hat{x}$), which is computed by integrating delayed and noisy sensory feedback ($x$) with internal prediction of the state. Importantly, feedback gains are tuned to task demands (*Liu and Todorov, 2007*; *Nashed et al., 2012*), and vary within a movement (*Dimitriou et al., 2013*). This task dependency gives rise to instances where all state variables ($\hat{x}$) are simultaneously available to the feedback controller, but some of them may not influence the motor output if deviations of the state vector in the corresponding dimension do not impact the performance.

We applied this control formalism to a task composed of a two-target sequence of reaching movements, such that two different sequences shared the first target, after which they diverged towards the second target (T1 and T2, *Figure 1C*). In the model, the estimated positions of the targets ($\widehat{T1}$, $\widehat{T2}$) are in the state estimate $\hat{x}$, hence the control law can be decomposed and expressed in the following equation (also see *Figure 2A*):

$$u(t) = k_1(t)\widehat{T1} + k_2(t)\widehat{T2} + k_x(t)\hat{x}(t)$$

where ($k_1$, $k_2$) are the feedback gains that determine how much and when each target ($\widehat{T1}$, $\widehat{T2}$) influences the motor output (*Figure 2A*). This formulation highlights that, even when both targets are simultaneously available to the feedback controller, the overlap between control gains ($k_1$ and $k_2$) will determine if and when the second target influences the first reach (*Figure 2B*). In the simulations, the control gains and the overlap between the two components of the sequence were not hardcoded but were found as an optimal control solution that improves the efficiency of the sequence as a whole (i.e. as a solution to holistic sequence planning in the sense that all targets are specified to the controller at once). For holistic sequence planning, we formulated the efficiency in terms of a cost function on the terminal movement errors and speed of reaching at both targets (T1 and T2), and total motor output across movement duration (see a model with terminal costs in Methods).

The model simulations revealed that, even though the sequence was planned as a whole, the necessity to slow-down at the first target, which was induced by an intermediate cost on speed at the corresponding time, influenced when the first reach became sensitive to the second target (*Figure 2C and D*), i.e., when the two reaches in the sequence coarticulate. When there was no constraint on speed at the first target, the simulated hand reached the first target with a high velocity (*Figure 2C*, left panel), and subsequently continued towards the second target at a high initial speed of >0.2 m/s. In this rapid transfer scenario, the control gains showed a cross-over effect where the control gain associated with the second target (k2) was non-zero, albeit small, during the first reach, showing an early overlap with the feedback gain (k1) (*Figure 2C*, middle panel). As a result, the location of the second target influenced the lateral hand deviation of the first reach (*Figure 2C*, right panel). The first reach was thus different depending on the location of the second target. Notably, the first reach was similarly influenced by the second target even in a different model where temporal buildup costs were considered instead of terminal costs (see temporal buildup cost model in Methods), to penalize the hand deviation from the targets across the duration of the sequence (*Figure 2—figure supplement 1A*).

Because the control gains also determine the response to external perturbations, the trajectories following external loads applied to the mass were also dependent on the second target (*Figure 2—figure supplement 2A*). Interestingly, the lateral deviation along the perturbed direction was smaller when the second target was in the direction of perturbation. Although this may appear counterintuitive, the model shows that it is an efficient solution, because a higher acceleration linked to reversal at T1, when the perturbation is in the direction of T2, incurs a higher motor cost. Hence, a smaller

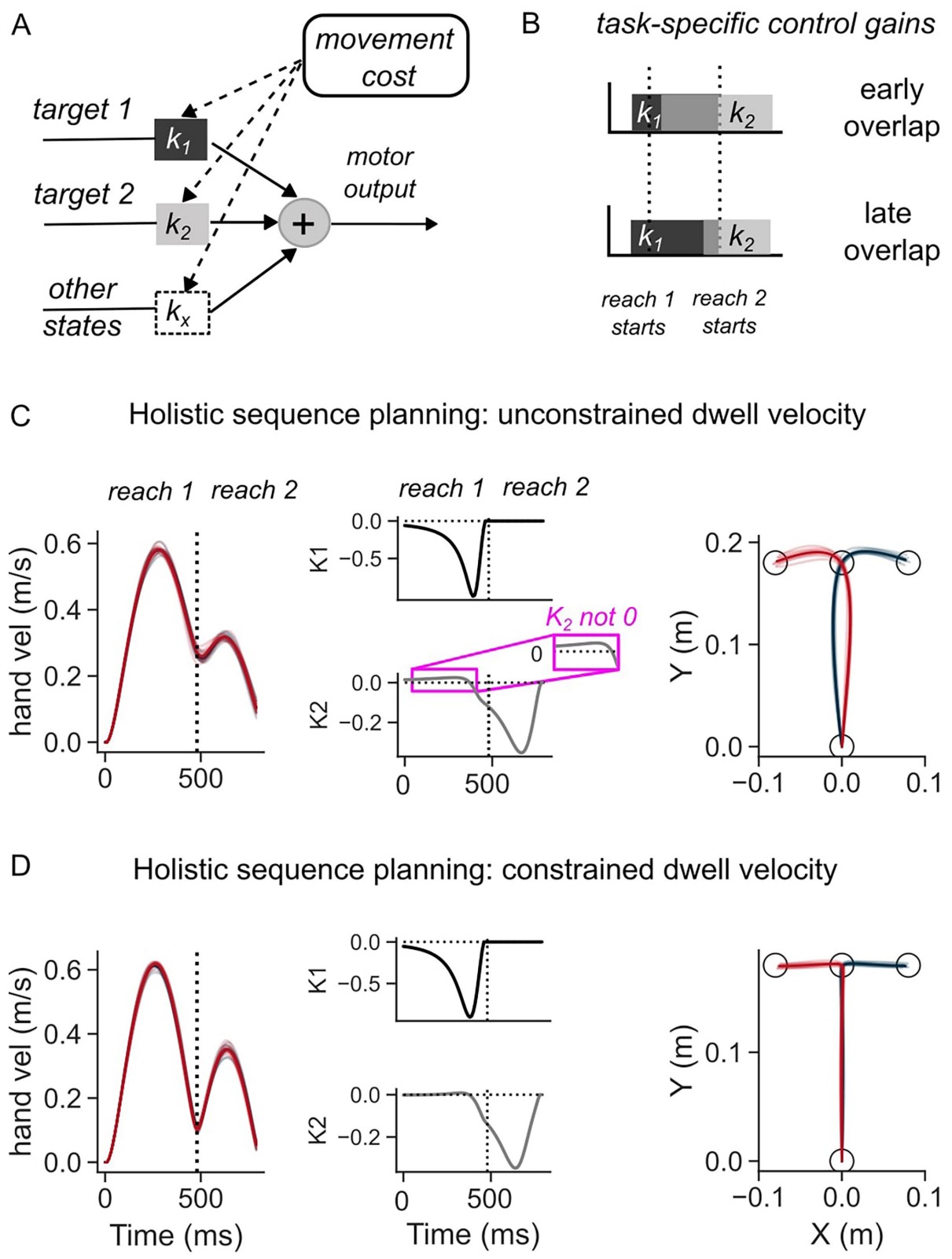

**Figure 2.** Simulations of the optimal feedback control (OFC) model in two-reaches sequence task. (**A**) Schematic illustration of the feedback control policy. Targets influence motor output through control gains, which are themselves derived by minimizing movement costs. (**B**) Illustration of how overlap of control gains can explain coarticulated reaches (top panel), and late influence of second target on first reach (middle). (**C**) Hand speed (left),

*Figure 2 continued on next page*

*Figure 2 continued*

control gains corresponding to the two targets (middle), and hand path (middle) when dwell velocity at target 1 was not constrained. (**D**) Same as (**C**) except the dwell velocity was penalized.

The online version of this article includes the following figure supplement(s) for figure 2:

**Figure supplement 1.** Simulations of the optimal feedback control (OFC) model in two-reaches sequence tasks using temporal buildup costs.

**Figure supplement 2.** Simulated perturbation trials when the whole sequence was planned for with no constraints on dwell velocity at the first target.

**Figure supplement 3.** Simulated perturbation trials when the whole sequence was planned for but with a constraint on dwell velocity at the first target.

**Figure supplement 4.** Simulations with two different values of intermediate (dwell) velocity cost parameters.

hand deviation is necessary to avoid high acceleration at T1. This result qualitatively persisted across broad variations in parameters related to the costs, delays, and point-mass dynamics (not shown). A similar result was observed when we used temporal buildup of kinematic error in the cost function (*Figure 2—figure supplement 2B*), instead of using the simple terminal costs, matching human data closely (see the experimental results).

On the contrary, the requirement to slow down at the first target, captured by an increased cost on speed, reduced the velocity at the end of the first reach and altered the shape of the trajectories (*Figure 2D*, left panel) compared to the rapid transfer scenario (*Figure 2C*, left panel). The control gain corresponding to the second target became non-zero at a later time, at the end of the first reach (*Figure 2D*, middle panel). Due to the later increase in this gain value, the influence of the second target on the motor output during the first reach was much reduced, hence generating a similar hand path to reach the first target across the two sequences (*Figure 2D*, right panel). Similarly, feedback responses to lateral perturbations that were applied early-on in the first reach were not dependent on the location of the second target (*Figure 2—figure supplement 3A*). The amount of lateral deviation when an unexpected perturbation was applied did not vary as a function of the direction of the second target. Qualitatively similar influence of the need to slow down at the first target on the trajectory of first reach was observed when temporal buildup costs were included instead of terminal errors (*Figure 2—figure supplement 1B* and *Figure 2—figure supplement 3B*).

It is worth noting that the OFC model can be generalized to longer sequences (*Kashefi et al., 2024*) through the incorporation of additional cost terms (in *Equation 10* of Methods) and targets, enabling simultaneous planning for more than two targets. Simulations of a sample three-reach sequence (*Figure 2—figure supplement 4*) revealed that, varying the cost of dwell velocity at intermediate targets ($w_2$ and $w_3$ parameters in Methods) caused a variation in control gains. Different amounts of change in control gains can be expected for intermediate versus late targets (*Figure 2—figure supplement 4A*). Notably, even when we used the same dwell velocity cost ($w_2 = w_3 = 0$), the observed velocity profiles were different between the two sequences towards different final targets (T3 up and T3 down) (*Figure 2—figure supplement 4B*). We tested a condition in which both sequence reaches were forced to have similar dwell velocity profiles by increasing the dwell velocity costs in the sequence towards one of the targets (T3 down), while leaving this parameter unchanged for the other target (T3 up). In this scenario, T3 up sequence had the parameters ($w_2$, $w_3$) = (0, 0), while T3 down sequence had the parameters (0.8, 0.8). In this case, the curvature of the first reach was different, and predominantly occurred due to differences in $K_2$ between the two sequence reaches (*Figure 2—figure supplement 4C*). These simulations highlight that, planning for a longer horizon sequence can indirectly influence the curvature of early reaches, due to the interaction between intermediate dwell constraints, spatial arrangement of targets, and sequence horizon in a task-dependent manner.

Overall, the model predicted that even if a feedback control policy was computed by optimizing the whole sequence over a long time-horizon, the requirements associated with intermediate goals determine how early in the sequence the second (future) target can influence the feedback controller (compare *Figure 2C* and *Figure 2D*; *Figure 2—figure supplement 1A* and *Figure 2—figure supplement 1B*; *Figure 2—figure supplement 2* and *Figure 2—figure supplement 3*).

## Correspondence between human behavior and model predictions

The experiments were designed to verify whether task requirements at the intermediate goals influenced hand trajectories and feedback control in a way that was expected from the model simulations. The first experiments tested the scenario in which there was no penalty on velocity during

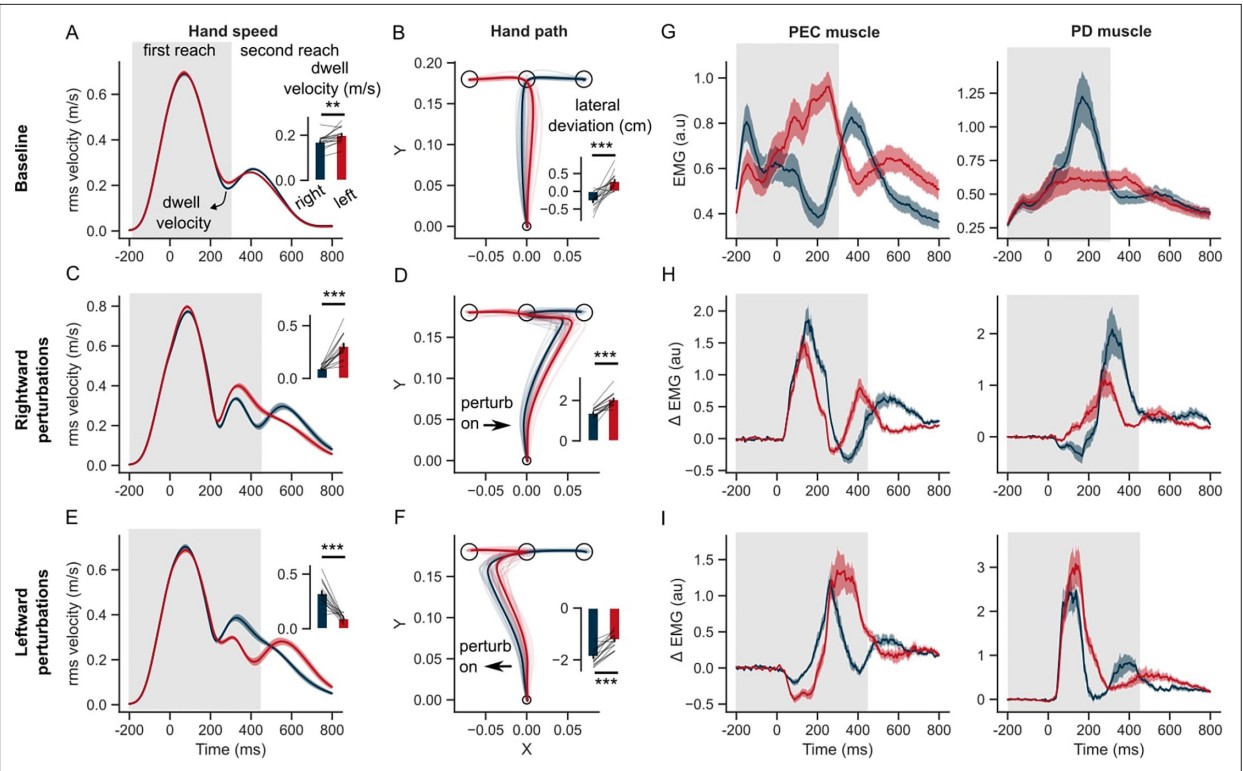

**Figure 3.** Hand kinematics in the GO task (experiment 1). (**A**) Hand speed and (**B**) hand path in unperturbed baseline trials. (**C, E**) Same as (**A**) but for perturbation trials. (**D, F**) Same as (**B**) but for perturbation trials. (**G**) Baseline EMG response of pectoralis major (PEC) and posterior deltoid (PD) muscles. (**H**) Perturbation response (baseline subtracted) of PEC and PD muscle during rightward and (**I**) leftward perturbations. Light curves in the hand path panels are the average paths of individual participants. Shaded area around the average curves represent the standard error (±sem) across participants around the mean. Gray colored box indicates the average duration of the first reach. Red and blue colors correspond to reaches made towards left and right second targets, respectively. t=0 ms represents the time of perturbation onset. *p<0.05, **p<0.005, ***p<0.001.

The online version of this article includes the following figure supplement(s) for figure 3:

**Figure supplement 1.** Comparison between first and last (eighth) unperturbed trials of a block, averaged across blocks of each participant in the GO task (Experiment 1).

the stop-over at the first target, referred to as the GO task. We instructed 15 human participants to perform a two-target sequence of reaching movements (*Figure 1C–F*). Both targets in a given sequence were displayed from the beginning of the trial, allowing for the preparation of both reaches. After receiving a GO cue (500–1000 ms after target display), participants were required to reach the first target within a prescribed time (500–650 ms), and then to proceed to the second target. There was no instruction to stop at the first target. To encourage fast transfer between the two reaches, the participants were allowed to remain in the first target for a maximum of 150 ms after a successful reach, before initiating the second reach, otherwise the second target disappeared. To determine if the feedback controller was tuned to the location of the second target, mechanical loads (spanning 200ms width) in randomized direction (rightward or leftward) were applied randomly in 20% of the trials soon after the hand left the starting location (4.5 cm) in forward direction from the starting location (this occurred typically at ~200 ms after the movement onset).

The model prediction about the influence of the second target during the first reach was clearly borne out of the experimental results. *Figure 3A* illustrates the average hand speed across baseline trials in the rapid transfer scenario. It is clear that the hand did not come to rest at the end of the first reach, but continued towards the second target at a relatively high transfer speed of ~0.2 m/s. The hand speed at the first target was slightly different between the 2 s target conditions (*Figure 3A*, inset; paired t-test: t(14) = –3.8, p=0.002). In line with model predictions, in the rapid transfer scenario, the location of the second target influenced the hand path during the first

reach (*Figure 3B*). Statistically, across trials and participants, lateral hand deviation during the first reach was significantly different between the two sequences (*Figure 3B*, inset; paired t-test: t(14) = −5.9, p<10⁻⁴).

Note that the experiments involve a block design where a sequence direction was maintained for 10 trials (with two randomly interleaved rightward and leftward perturbation trials), while the blocks themselves were randomly interleaved (*Figure 1E*). To test whether the influence of the second target on hand deviation was due to trials occurring late within a block, we computed the hand path in the first and last trials and averaged across all blocks for each participant. Group data in *Figure 3—figure supplement 1* illustrate that the hand deviation was significantly different between the leftward and rightward sequence conditions from the very first trial within a block (paired t-test: t(14) = 5.7, p<10⁻³). Similarly, the hand deviation was significantly different between leftward and rightward sequences in the last trial within a block (paired t-test: t(14) = 5.9, p<10⁻³). Additionally, a temporal effect was observed in the leftward sequence condition, where the last trial incurred a higher hand deviation compared to the first trial within blocks (paired t-test: t(14) = 3.9, p=0.003). Note that the above results include correcting for multiple comparisons using the Holm-Bonferroni method. Overall, the block design of the experiment may lead to small differences between the first and last trials within a block, but the effect of the second target on the lateral hand deviation was present from the very first trial within a block, suggesting that qualitatively similar results can be expected in experiments without a block design.

During perturbation trials, hand trajectories during the first reach were influenced by the location of the second target. In rightward perturbation trials, the dwell velocity and hand deviation (*Figure 3C and D*) were significantly different across the 2 s target conditions (paired t-test: t(14) = −7.1, p<10⁻⁵ for dwell velocity; t(14) = −7.9, p<10⁻⁵ for hand deviation). Similar differences across the two sequences were observed in the leftward perturbation trials (*Figure 3E and F*) (paired t-test: t(14) = 6.9, p<10⁻⁵ for dwell velocity; t(14) = −8.2, p<10⁻⁶ for hand deviation). The influence of the second target on the lateral hand deviation was qualitatively similar to that observed in model simulations, and counterintuitive to what we might expect without the help of the model simulations. As observed in the model simulations (see also *Figure 2—figure supplement 2*), lateral hand deviation was smaller when the perturbation was in the direction of the second target (T2) and vice-versa. This was consistent for both rightward and leftward perturbation conditions. Both the model and humans expressed this strategy that can be seen as an emergent feature of efficient feedback control during the production of movement sequences. Additionally, even though behavior was reproduced in simulations, changing the cost of control effort and/or accuracy of intermediate reaches could modulate the sequence-dependent changes in curvature.

We identified a pair of muscles in the upper-arm shoulder joint (Pectoralis Major, PEC, and Posterior Deltoid, PD) that were strongly recruited by the lateral perturbation (see earlier studies with a similar experimental setup *De Comite et al., 2022*; *Crevecoeur et al., 2019*). These muscles also steer the hand toward the first and second targets in unperturbed conditions. PEC muscle displayed agonist burst coinciding with the acceleration of the hand towards the first target, while the PD muscle provided the antagonist burst to decelerate the hand near the first target. Notably, both muscles displayed a clear difference in baseline EMG activity during the first reach, depending on whether the following reach involved right or left second targets (*Figure 3G*). Following the perturbation (left or right), the stretch responses were visibly larger in both PEC and PD muscles compared to the baseline response (*Figure 3H, I*). The perturbation response in the stretched muscle – PEC for the rightward load (*Figure 3H*, left panel) and PD muscle for the leftward load (*Figure 3I*, right panel) – was visibly larger if the second reach involved movement reversal compared to the first reach where the perturbation was in the direction of the second target. Interestingly, even the unloading response (below baseline) in the unstretched (shortened) muscle – PD for the rightward load (*Figure 3H*, right panel) and PEC for the leftward load (*Figure 3I*, left panel) – was larger if the second reach involved direction reversal. Overall, the perturbation responses – both stretch and shortening muscle responses – indicated vigorous feedback corrections against the load when the second target induced movement reversal at the first target, explaining the smaller lateral deviation of the hand when the perturbation was in the direction of the second target.

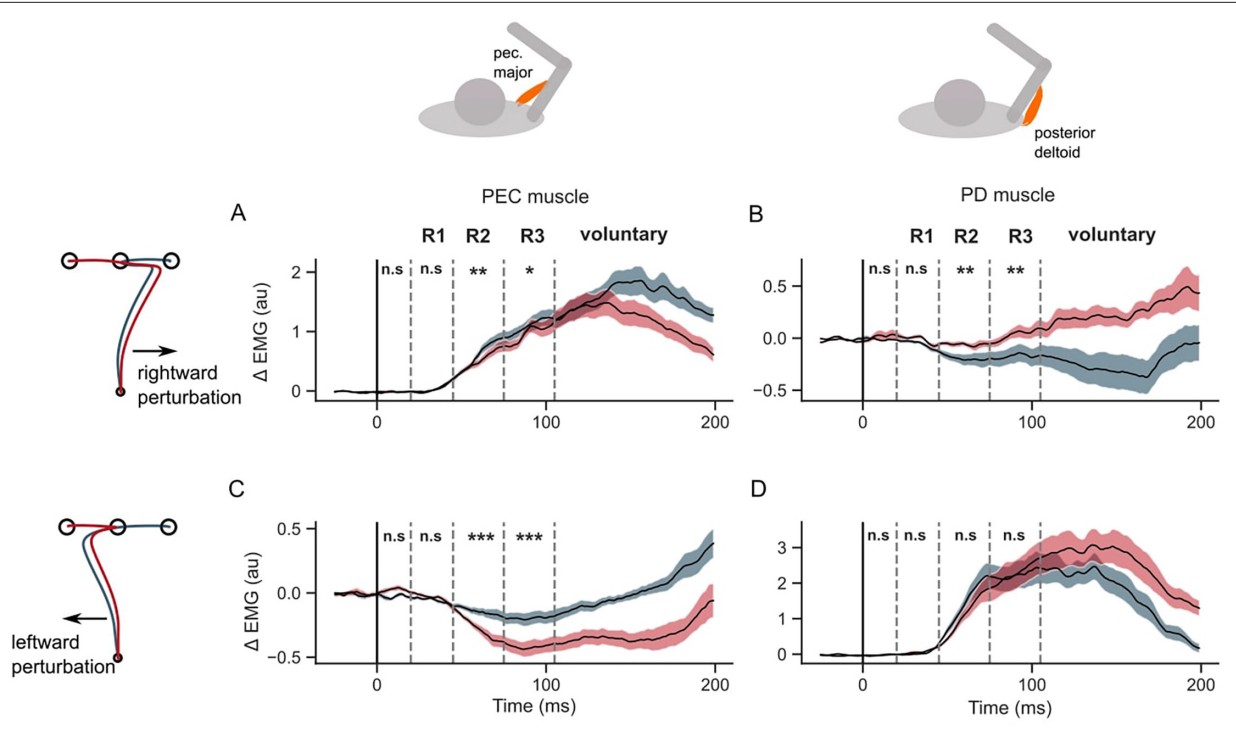

**Figure 4.** EMG perturbation responses in the GO task (Experiment 1). (**A**) Pectorialis major (PEC) muscle response to a rightward perturbation. (**B**) Posterior deltoid (PD) muscle response to a rightward perturbation. (**C**) PEC muscle response to a leftward perturbation. (**D**) PD muscle response to a leftward perturbation. Red and blue colors represent sequences to leftward and rightward second targets, respectively. Shaded area represents the standard error (±1 SEM) across participants around mean. All traces were aligned to perturbation onset. t=0 ms indicates the time of perturbation onset. *p<0.05, **p<0.005, ***p<0.001 and n.s indicates non-significant effects of one-tailed t-test with p>0.05.

## Sequence-dependent long-latency feedback responses

We assessed when the muscle stretch and shortening responses showed dependency on the second target. If the two targets were simultaneously used to form an efficient feedback controller for the whole sequence, one would expect that the sensitivity of the response to the second target to become visible in the long-latency epoch, considering responses in this epoch as a proxy of flexible, state-feedback controller in the primate nervous system (*Kalidindi and Crevecoeur, 2023*; *Scott, 2016*). Overall, the EMG responses indicated a significant influence of the second target on the long latency responses in both stretched and shortened muscles (*Figure 4*). When perturbed in the rightward direction, the PEC muscle (agonist) response increased in both sequence conditions, which counters the perturbation by increasing the leftward pull. Given the lower hand deviation in the rightward sequence (*Figure 3D*), we asked if the reduced deviation was due to a larger leftward pull from the PEC muscle. Indeed the PEC stretch response was significantly larger in the rightward sequence than the leftward sequence in the long-latency epoch, in both R2 (one tail paired t-test: $t(14) = 3.5$, $P<0.002$) and R3 (one tail paired t-test: $t(14) = 2.25$, $P=0.02$). On the other hand, the PD muscle shortening response was significantly larger (below baseline) in the rightward sequence than the leftward sequence in both R2 (one-tailed paired t-test: $t(14) = -3.2$, $P=0.003$) and R3 epochs (one-tailed paired t-test: $t(14) = -3.5$, $P=0.002$).

When the perturbation load was to the left, the PEC shortening response (below baseline) was significantly larger in the leftward sequence compared to the rightward sequence in both R2 (one tail t-test: $t(14) = 7$, $P<10^{-5}$) and R3 epochs ($t(14) = 8.1$, $P<10^{-6}$). Such a large unloading response in the PEC muscle could facilitate a larger correction in the leftward sequence by reducing the leftward pull in the direction of the applied load. While the shortening response in PEC was larger, the stretch response in PD muscle was not significantly larger in the leftward sequence throughout the long-latency time period (one-tail paired t-test; R2 epoch: $t(14) = 2.8$, $P=0.99$; R3 epoch: $t(14) = -0.13$, $P=0.44$). On the contrary, the PD stretch response was significantly smaller in the leftward sequence

in the R2 epoch (one-tailed t-test: t(14) = 2.8, *P*=0.006). All muscles exhibited differences in the voluntary time window (120–180ms) during the first reach depending on the location of the second target (two-tailed paired t-test: t (14) absolute value between (3.5–7.4), and p-value between ($1e^{-6}$ – 0.012) across muscles and perturbation types).

Next, we verified that the differences reported above did not follow from mere differences in baseline muscle activities at perturbation onset. Muscle responses exhibit a property called 'automatic gain scaling' where the stretch response increases proportionally to the pre-perturbation muscle activity (*Pruszynski et al., 2009*). If gain-scaling was significant, then we expected to observe a difference in the R1 period response between the two sequence conditions. We did not observe any sequence-related modulation of the stretch responses in the R1 epoch, suggesting that any potential differences at baseline did not have a strong impact. It is also important to realize that the sequence direction that involved a larger stretch response in the corresponding muscle was always associated with lower levels of activity in the baseline condition, thus potential biases induced by baseline differences were in the direction opposite to the effect reported above.

Overall, the results indicate, the when required to perform movement sequences rapidly, the sensorimotor circuits that mediate long latency feedback responses showed modulation that was dependent on the secondary target. The specificity of the controller to the second target was evident when perturbations were applied early-on during the first reach. These observations agree with the predictions of the feedback control model that, with low or absent penalty on the velocity at the first target, the second target influences the hand path and feedback responses to perturbations prior to the first target.

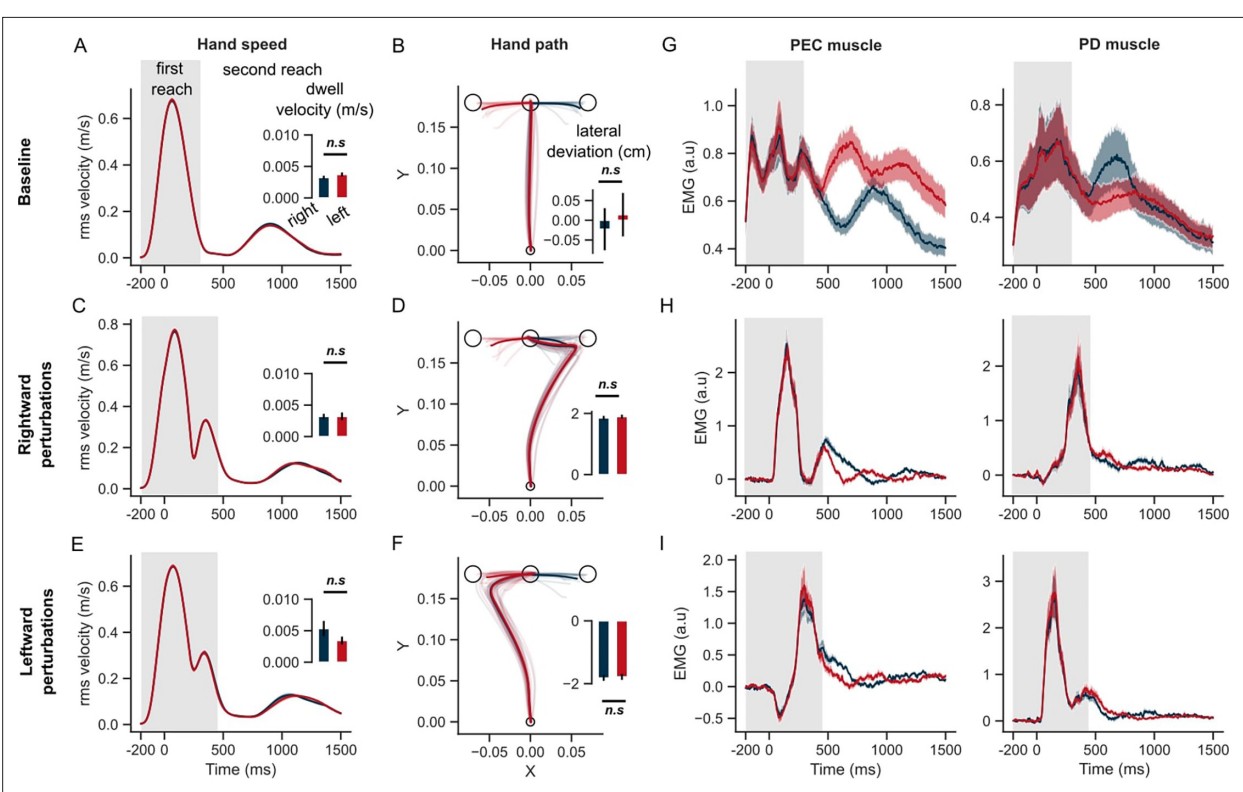

**Figure 5.** Hand kinematics in the STOP task (experiment 2). (**A**) Hand speed and (**B**) hand path in unperturbed baseline trials. (**C, E**) Same as (**A**) but for perturbation trials. (**D, F**) Same as (**B**) but for perturbed trials. (**G**) Baseline EMG response of pectorialis major (PEC) and posterior deltoid (PD) muscles. (**H**) Perturbation response (baseline subtracted) of PEC and PD muscle during rightward and (**I**) leftward perturbations. Light curves in hand path panels are average paths of individual participants. Shaded area around the average curves represent the standard error (±sem) across participants around the mean. Gray colored box indicates the average duration of the first reach. Red and blue colors correspond to reaches made towards left and right second targets, respectively. t=0 ms represents the time of perturbation onset. *p<0.05, **p<0.005, ***p<0.001.

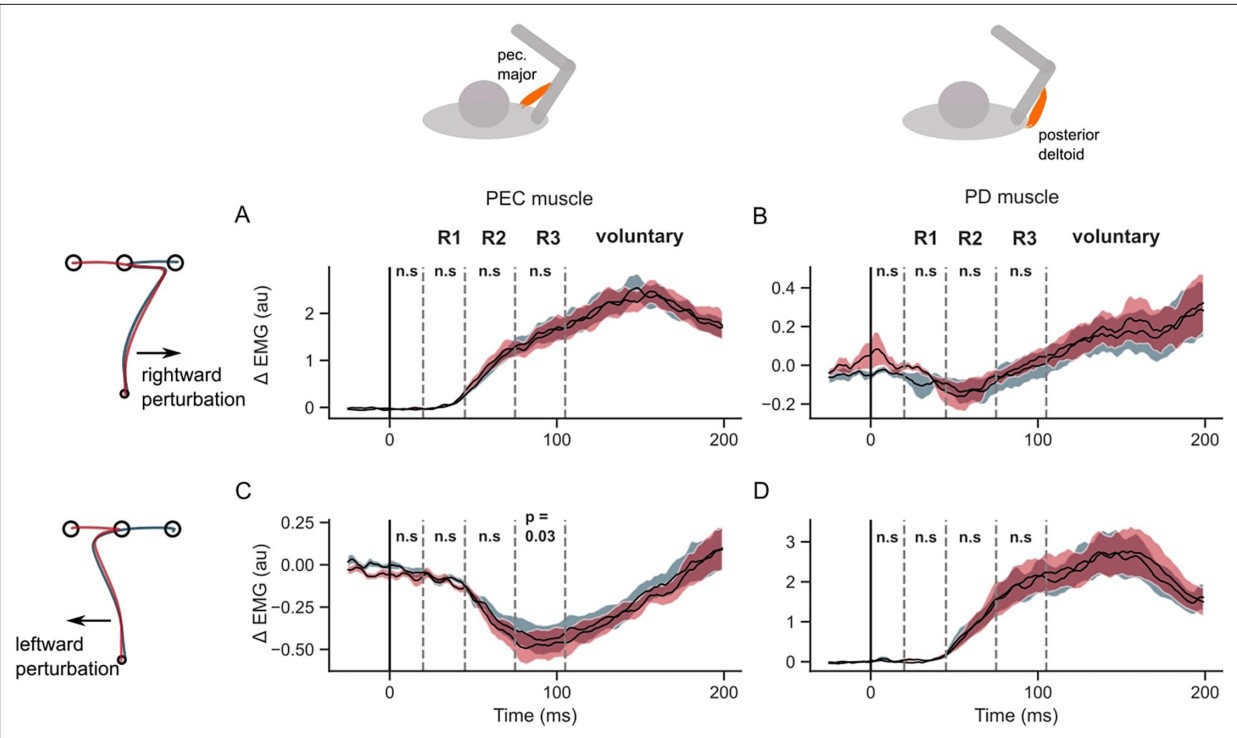

**Figure 6.** EMG perturbation responses in the STOP task (Experiment 2). (**A**) Pectorialis major (PEC) muscle response to a rightward perturbation. (**B**) Posterior deltoid (PD) muscle response to a rightward perturbation. (**C**) PEC muscle response to a leftward perturbation. (**D**) PD muscle response to a leftward perturbation. Red and blue colors represent sequences to leftward and rightward second targets, respectively. Shaded area represents the standard error (±1 SEM) across participants around mean. All traces were aligned to perturbation onset. t=0 ms indicates the time of perturbation onset. n.s indicates non-significant effects of one-tailed t-test with p>0.05.

## Separation of Sequence Elements

Another clear prediction of the model is that when there is a cost on velocity at the first target, the first reach should be less dependent on the second target. To test this prediction experimentally, we performed a STOP task where 14 subjects were asked to perform the same two-target reaching movements with a different speed constraint (**Figure 1F**). They were required to bring the hand to a near-complete halt (speed <0.5 cm/s) before starting the second reach.

**Figure 5** illustrates that the hand speed at the first target dropped to near zero levels in all unperturbed, leftward, and rightward perturb conditions. Clearly, there was no influence of the second target on the velocity of reaching the first target (two-tailed paired t-test: t(13) = –1.9, *P*=0.08 in the baseline condition; t(13) = –0.01, *P*=0.99 for rightward perturbations; t(13) = 2.1, *P*=0.055 for leftward perturbations). The hand deviation during the first reach was similar between the two sequences (two-tailed paired t-test: t(13) = –1.8, *P*=0.08 in the baseline condition; t(13) = –1.2, *P*=0.26 for rightward perturbations; t(13) = –1.2, *P*=0.25 for leftward perturbations). Moreover, the unperturbed EMG responses of the PEC and PD muscles were similar throughout the first reach, and diverged only after the end of the first reach, while moving toward the second targets (T2). In rightward and leftward perturbation conditions, the perturbation responses (*ΔEMG*) of the PEC and PD muscles (**Figure 5D and E**, and **Figure 6**) were not statistically different pre-perturbation, R1, R2, or R3 epochs. Only the PEC perturbation response, to a leftward perturbation, showed a weak effect of the second target location (R2 epoch: t(13) = 2.1, p=0.055; R3 epoch: t(13) = 2.44, p=0.03). To conclude, these results suggested that the second target did not influence the first reach when the hand was required to slow-down at the first target.

A straightforward interpretation could be that the stopping at the first target invoked a completely different strategy in which the control of the two reaches was performed independently (**Figure 1A**), effectively separating the two movements, whereas executing them rapidly could produce the merging of the two sub-movements into a coarticulated sequence. While this is conceptually valid, it

is not necessary and the model provides a more nuanced view: both apparent separation or coarticulation of the two motor patterns can be explained within the same framework of flexible feedback control. These different modes of sequence execution still require proper specification of the task constraints in the model, such as a number of intermediate steps, dwell-time, or velocity limit. Such specifications must be considered as input to the controller.

## Discussion

The results presented here are consistent with recent behavioral studies showing that the information about future targets in a sequential task is integrated at the stage of motor planning (*Diamond et al., 2017*; *Gottwald et al., 2018*; *Kashefi et al., 2024*; *Ariani et al., 2021*; *Engel et al., 1997*). Our key contribution is twofold: we showed from computational and behavioral perspectives that the coarticulation and separation of sequence components depended on task instructions, and likely reflected efficient solutions of the same continuous control problem with distinct penalties at the intermediate goals; second, the results highlighted that long-latency feedback control was modulated by future targets in a coarticulated sequence, consistent with theories suggesting that movement planning is synonymous with derivation of a control policy (*Crevecoeur et al., 2014*; *Wong et al., 2015*).

A key question the model and experiments addressed was whether the integration of multiple goals required processing in upstream voluntary circuits, indicating timed, voluntary recall of future goals (*Zimnik and Churchland, 2021*; *Schimel et al., 2023*; *Logiaco et al., 2021*). Instead, our results support the idea of flexible and simultaneous processing of sequence elements, where a low-level controller plays a crucial role in combining these elements. Our model based on optimal feedback control showed that even if two reaches are planned as a whole, the second target does not necessarily influence the controller early. The constraints placed on the swiftness of transition around the first target modulated when the second target started to influence the sequence. This was clearly apparent in the hand deviation during the first reach, and further revealed by perturbing participants' hands with external loads, evoking sequence-dependent long-latency responses in Experiment 1. Consistent with this model (*Figure 1B*), recently an artificial neural network model with simultaneous processing of targets reproduced the population activity of the primary motor cortex (see *Wang et al., 2024*), a region that contributes significantly to long latency feedback responses (*Scott, 2016*; *Pruszynski et al., 2011*). Together, these studies point to the importance of considering feedback gains, and suggest a special role of transcortical feedback through M1 in the production of movement sequences. Notably, in the framework of optimal feedback control, an intermediate goal is equivalent to a via-point that constrains the execution of the sequence (similar to *Ramkumar et al., 2016*). It is thus possible that coarticulation in motor systems be processed similarly as other kinds of movement constraints, such as via-points, avoiding obstacles, or changes in control policies.

Although OFC has been predominantly used as a behavioral-level framework agnostic to neural activity patterns, it can shed light on the planning, state estimation, and execution-related computations in the transcortical feedback pathway (*Takei et al., 2021*). Using OFC, our study proposes a novel and precise definition of the difference to expect in neural activities in order to identify coarticulated versus independent sequence representations from a computational point of view. Because each condition (i.e. overlapping versus non-overlapping controllers as in *Figure 2*) was associated with different cost-functions and time-varying control gains, it is the process of deriving these control gains, using the internal representation of the task structure, that may differ across coarticulated and separated sequence conditions. To our knowledge, how and where this operation is performed is unknown. A corollary of this definition is that the preparatory activity (*Zimnik and Churchland, 2021*; *Ames et al., 2019*) may not discern independently planned or coarticulated sequences because these situations imply different control policies (and cost functions), as opposed to different initial states. Moreover, the nature of the sequence representation is potentially not dissociable from its execution for the same reason.

Many studies highlight a phenomenon called chunking, where a subset of individual elements of a sequence are concatenated and represented as a unified whole called chunks (*Ramkumar et al., 2016*; *Verwey, 2001*; *Bera et al., 2021*). In the pursuit of finding motor chunks, recent neural recordings explored for holistic (or) unified representations in the circuits that produce motor commands, and found no such representations (*Wong and Krakauer, 2019*; *Yokoi and Diedrichsen, 2019*; *Zimnik and Churchland, 2021*; *Berlot et al., 2020*). Hence it was suggested that chunking resulted from a

cognitive or high-level recall of intermediate goals sent to the controller for independent execution (*Wong and Krakauer, 2019*, see *Figure 1A*, but, see *Huberdeau et al., 2023*). On the contrary, our developments also provide a computational definition of motor chunks. From our model, motor chunking (or a unified representation) does not necessarily imply a single (unified) preparatory period signal representing multiple goals. Motor chunking may represent the formation of a composite cost function, hence a composite control policy, that can jointly optimize multiple sub-movements (see *Figure 1B*). In this view, motor chunking results from an acquired motor ability to form overlapping control policies for contiguous sequence elements (*Figure 1B* and *Figure 2*). Importantly this property depends on the instruction at intermediate goals, thus even a motor chunk may falsely appear to be independently controlled (*Figure 1A*), masking the signature of sequence representation in neural data even if the controller was formed at once by considering multiple goals.

There are differences between the model and human movements that require further study, such as the influence of sequence direction on the dwell velocity, which was not observed in the model simulations (compare velocity plots in *Figure 2C* and *Figure 3A*). A limitation is that nonlinear dynamics were ignored in our model. However, the potential impact of non-linear mechanics on limb trajectory could be completely suppressed by participants in Experiment 2, since we observed a straight reach path, thus making the curvature observed in Experiment 1 more likely related to the task constraints rather than the biomechanics which did not differ between experiments. Yet, the quantification of how biomechanical factors influence control may deserve a dedicated work, as it has been shown that biomechanics can also influence coarticulation in a speech production task (*Ostry et al., 1996*). Despite the limitations, the simple linearized OFC models were crucial in determining the relationship between three properties of the task demands, its impact on coarticulation, and feedback control. Our approach is consistent with how ideal actor/estimator models have been predominantly employed as normative models in formalizing an initial hypothesis, and simulating these to generate testable predictions (*De Comite et al., 2022*; *Liu and Todorov, 2007*; *Cesonis and Franklin, 2021*; *Nashed et al., 2012*; *Crevecoeur et al., 2019*; *Takei et al., 2021*; *Diedrichsen, 2007*; *Crevecoeur et al., 2020b*; *Izawa et al., 2008*; *Ikegami et al., 2021*; *De Comite et al., 2023*; *Mistry et al., 2013*). Such normative models should be distinguished from detailed biomechanical models that can trade-off interpretability and predictability for achieving better quantitative fits. Furthermore, the main feature of the model – to generate a task-dependent motor output as a combination of state feedback – is not limited to linear controllers and linear body dynamics, but is applicable even when the control policy is non-linear (see *Kalidindi et al., 2021*; *Lillicrap and Scott, 2013*).

The modeling and empirical results presented here describe the computations underlying a two-reaches sequence. Although two is admittedly the lowest possible length of a non-trivial sequence, it enables comparison with similar tasks that have been used in recent neurophysiological studies (*Zimnik and Churchland, 2021*; *Ben-Shaul et al., 2004*; *Hatsopoulos et al., 2003*; *Wang et al., 2024*) to understand the neural computations underlying the control of sequences. We expect that longer sequences be formed in a similar way as we showed that there is no theoretical constraint on the number of intermediate goals in the model and the addition of sequence elements is straightforward. Thus, we expect the combination of two elements can be the basis of longer sequence formation constructed on the same principles.

Besides the implication of our computational model of co-articulation, it has been often documented that swift execution of sequences was achieved by habituation, where a set of sequences are practiced for several days to months (*Ramkumar et al., 2016*; *Yokoi and Diedrichsen, 2019*; *Zimnik and Churchland, 2021*; *Lu and Ashe, 2005*; *Berlot et al., 2020*; *Huberdeau et al., 2023*; *Ariani and Diedrichsen, 2019*; *Maceira-Elvira et al., 2022*). Such a process potentially involves a slow time-scale linked to the acquisition of a motor skill (*Maceira-Elvira et al., 2022*; *Beukema and Verstynen, 2018*), which differs from the results presented here, as the participants were able to execute accurate sequences after a pre-training of around 100 repetitions. Here, the participants received explicit information about the dwell constraints at the intermediate goal, resulting in quick coarticulation (also see *Sosnik et al., 2004*; *Kashefi et al., 2024*; *Friedman and Korman, 2019*; *Sosnik et al., 2007*; *Sporn et al., 2020*). This is in line with task-dependent modulation of existing reaching controllers, which can be rapidly switched from one trial to another (*De Comite et al., 2022*; *Cesonis and Franklin, 2022*). This suggests that there exist several mechanisms operating at different time-scales and responsible for a combination of sequence elements. One mechanism is a slow habituation process in which the

subject learns de novo to build a novel controller for different elements in the sequence (*Huberdeau et al., 2023*; *Yang et al., 2021*), whereas the other mechanism is a fast modulation of this consolidated controller by task demands. In principle, the two mechanisms are dissociable with perturbations, which may allow future studies to investigate how they interact and to identify which brain regions mediate these components of skilled motor behavior.

## Materials and methods
### Experiment design

A total of 29 volunteers (11 men, 18 women, aged between 20 and 35 years) participated in two experiments (n=15 with 9 women in Experiment 1; and n=14 with 9 women in Experiment 2). All participants were neurologically healthy, and right-handed and gave their informed consent according to a protocol approved by the ethics board at UCLouvain. Experiments did not exceed 2 hr, and participants were compensated for their time.

The experiments used a robotic device (endpoint KINARM, BKIN Technologies), permitting hand movement in the horizontal plane. In addition to recording the hand kinematics and forces on the robot handle, the KINARM can perturb the arm by applying mechanical loads at the handle. Projected targets (circles of chosen diameters) and hand feedback (white dot, 0.5 cm diameter) were presented to the participant from a heads-up display consisting of an overhead projector and a semitransparent mirror. During the experiment, direct vision of the arm and hand was occluded by a metal partition.

Participants were instructed to grasp a robotic handle and move the hand-aligned cursor to the home target (radius 0.6 cm). The home target was initially filled with red color but turned green when the participant's cursor entered the home target. Two targets (open circles with white edges) appeared simultaneously once the participants entered the home location. The first target was situated 18 cm in the forward direction on the midline, while the second target was either situated to the left or right from the midline, displaced laterally from the first target by 8 cm (*Figure 1C*). Hence, at each trial, the participant could simultaneously view the first and second targets, T1 and T2, respectively. After holding the hand within the home location for 500–1000 ms ('preparatory period' uniformly distributed), the two targets were filled with red color from open circles, acting as a GO cue, indicating the participant to start executing the sequence. As the experiment involved many movements, in order to reduce the possibility that participants missed a GO cue, a 50 ms beep sound was presented simultaneously as an additional, auditory cue.

### Experiment 1

Participants (n=15) were instructed to initiate the sequence as soon as they received the GO cues. For the first reach towards T1, participants were asked to move as straight and as quickly as possible. The first reach was deemed successful if T1 was reached between 500–650 ms after receiving the GO cue. Upon successful first reach, the circle corresponding to the first target turned green. As soon as the first target was successfully reached, a counter of 150 ms was initiated (unknown to the participant), leading to the disappearance of the second target if the hand did not leave the first target quickly. The second target remained in red color, if the hand left the first target within 150 ms, and subsequently turned green once the hand reached T2. If participants successfully completed the sequence by acquiring T1 and T2, and remained in T2 for at least 1 s, then they received two points. If they successfully acquired T1 but failed to quickly leave T1 (<150ms after reaching) or to reach T2, then they received one point. The score was projected on the screen and was updated only after the end of the trial and not during any reaching or preparatory period. The short latency of dwelling (at most 150 ms) in the first target encouraged the participants to swiftly move towards T2 soon after T1 was acquired. If the hand reached T1 in <500 ms after the GO signals, then both T1 and T2 disappeared indicating that movement was too fast, leading to failure of this trial. If the hand reached T1 in >650 ms after the GO signals, then the T1 circle never turned red, indicating that the trial was too long. In both cases where T1 is not acquired properly, the participants did not receive any points even if they reach T2. These cases of failed trials due to movements that were too fast or too slow were included to encourage low variation in movement durations, but all trials were used for analysis. Participants were asked to try to complete the sequence by reaching T2, even if they feel that T1 was not reached within the prescribed time.

In a subset of random trials (20% of the total number of trials), a mechanical load, of ±7 N in magnitude with a 10ms linear rise time and a 200 ms width, was applied randomly in the lateral direction (*x*-axis in *Figure 1*), once the hand reached 1/4th of the forward distance between the home location and the first target (T1). Notably, in perturbation trials, the successful reaching time to T1 was set to be between 500–800 ms, in contrast to 500–650 ms used in unperturbed trials. This additional time, not informed to the participant, was necessary to account for the increased path length induced by the perturbation. This ensured a good proportion of perturbation trials with complete sequence execution, instead of the participants' not attempting the reach towards T2 due to failure of reaching T1 properly.

Participants were given the following instructions verbally: 'Wait in the starting circle until you receive a GO signal, where the target circles turn red and you will simultaneously hear a beep sound. When the circles turn red, react quickly, move as soon, and as straight as possible to target 1 and then move to target 2. You will get two points at the end of the trial if you reach T1 in the prescribed time window and then move to T2, and in all other cases, you will not receive any points. Importantly, once you reach T1 you should try to come out of it quickly. If you stay in T1 for more than 150ms then T2 will disappear and you will receive only one point. Additionally, in some trials, a force will perturb your hand towards the right or left direction randomly while moving towards T1. The instructions remain the same in the presence of perturbations. Try to score as many points as you can'.

## Experiment 2

Participants (n=14) performed two-target reaching sequences similar to Experiment 1 except that they were required to slow-down while reaching the first target (T1). Similar to Experiment 1, two targets were simultaneously displayed (as open circles) once the hand enters home location. At the GO signals, only the first target turned red, while T2 remained as an open circle. Participants were asked to reach T1 as straight and quickly as possible, between 500–650 ms, but importantly to slow-down at T1 before continuing towards T2. In this experiment, the condition for the target T2 to turn red, indicating the participant to move towards it, was that the hand velocity had to fall below a threshold of 0.5 cm/s. The scores were incremented in the same way as in Experiment 1.

## Block design (randomized within a set)

The experiments consisted of nine sets, where each set was composed of 10 blocks equally divided between rightward and leftward sequences (i.e. five blocks with rightward and five blocks with leftward second target location relative to T1) (see *Figure 1*). Notably, the rightward and leftward blocks within any given set appeared in a pseudo-random order. Each block is composed of 10 two-reach sequence trials. Hence, a total of 900 trials were performed across nine sets (9 sets ×10 blocks/set × 10 trials/block = 900 trials). After each set, participants were given a 30 s to 1 min break depending on their willingness to continue to the next set. Across the 10 trials within a block, the location of the second target (T2) remained constant (either to the right or to the left of T1). Within each block of 10 trials, two random trials involved perturbations (one rightward and one leftward perturbation), while the remaining eight trials were unperturbed. Hence, each perturbed direction included an equal number of rightward and leftward sequences, allowing us to examine the influence of the location of the second target on the perturbation responses in each direction of perturbation.

## Pre-experiment training

The participants were first trained to execute single reaches to reach and stop at T1, without presenting T2, for 50 practice trials. Then they performed a maximum of one set of two-reach sequences (10 blocks × 10 trials/block = 100 trials), without any perturbation trials. This allowed them to learn the task without the complexities associated with reacting to external perturbations. Finally, they performed one more set, including 10 blocks where each block is composed of eight unperturbed trials and two perturbation (rightward and leftward) trials. We added this pre-training before the main experiments to familiarize the participants with the sequence task. We observed that the scores of most participants improved by the end of this small pre-experiment training set in both experiments.

## Muscle recordings

We recorded the activity of mono-articular shoulder muscles using Delsys Bagnoli surface EMG setup: pectorialis major (PEC) and posterior deltoid (PD), involved in the production of lateral forces and known to be largely recruited following similar perturbations with the same setup (*Crevecoeur et al., 2020a*; *Mathew et al., 2020*; *De Comite et al., 2022*). The electrodes were attached to the skin above the muscle belly after slight abrasion with alcohol. The signals were amplified (gain: $10^4$), and digitally bandpass filtered with a dual-pass, fourth-order Butterworth filter (20–400 Hz bandpass). EMG data were then normalized to the average activity across 1 s recorded when the participants maintained postural control at the home target against a background load of 12 N applied three times in each direction. This calibration was performed at the beginning of the first and fifth blocks.

## Statistical analysis

### Behavior and kinematics

We extracted the hand deviation, and dwell velocity at the first target, as defined below.

### Lateral hand deviation

The average lateral hand deviation (in meters) from an imaginary straight-line connecting the home location (H) to the first target location (T1), during the first reach. This is a signed quantity where a positive/negative sign indicates hand deviation to the right/left laterally.

### Dwell velocity

The minimum root-mean-square velocity (of x and y dimension) in the duration in which the hand dwells near the first target – below 0.5 cm distance from the edge of T1.

Kinematic variables of each participant were aligned to the time of perturbation onset (t=0), and then averaged across trials for group data analysis. We performed paired t-tests (one-sampled and two-sampled), unless explicitly stated otherwise, to determine significant differences in the extracted kinematic variables.

### Perturbation-related muscle activity (ΔEMG)

Our primary interest was to compare different epochs of muscle activity to gain further insights into the sensitivity of feedback responses to the location of the second target, when the hand was perturbed early-on in the reaching towards the first target. The epochs of perturbation-related muscle activity of most interest were based on previous reports (*Kurtzer et al., 2008*; *Pruszynski et al., 2009*; *Pruszynski et al., 2011*) and were categorized temporally: baseline = –100–0 ms, R0=0–20 ms, R1=20–45 ms, R2=45–75 ms, R3=75–105 ms, and voluntary (Vol)=120–180 ms. Note that all EMG signals for each participant were aligned to the perturbation onset.

We were most interested in comparing how the feedback responses are influenced by the location of the second target, even though the perturbations were applied in the beginning of the first reach. Accordingly, we calculated the average EMG response of each participant in the perturbed trials and subtracted this from the average EMG responses across unperturbed trials for each block within a set. We refer to this quantity as perturbation-related muscle activity (ΔEMG). Paired t-tests were performed using group averages to determine if the location of the second target had any statistical effect on the perturbation-related muscle activity. It should be noted that corrections for multiple comparisons do not apply here for two reasons: first, the samples at each epoch are involved in only one comparison; second, consecutive samples are not statistically independent. If there is a significant difference at a given epoch, it is very likely that there will be a significant difference in the next epoch because signals do not vary instantaneously. Hence, the risk of false-positive must not be controlled.

## Model

The model describes the translation of a point-mass similar to the mass of an arm (m=2.5 kg) in the horizontal plane. The coordinates corresponded to the experiments, such that the first target was directly in the forward-reaching direction (along the y-dimension), and the second targets were at the same y-coordinate but displaced in the x-dimension by 8 cm to the right or to the left of the first target.

## Dynamics of the effector

The state-space representation of the point-mass (from *Crevecoeur et al., 2019*; *Crevecoeur et al., 2020b*), with mass 'm,' was as follows:

$$m\ddot{x} = -g\dot{x} + f_x + f_{LOAD} \tag{1}$$

$$m\ddot{y} = -g\dot{y} + f_y \tag{2}$$

$$\tau\dot{f}_x = u_x - f_x \tag{3}$$

$$\tau\dot{f}_y = u_y - f_y \tag{4}$$

Dot(s) correspond to time derivatives. The variables $f_x$ and $f_y$ are the forces applied by the controlled actuator to the mass, and these forces are a first-order response to the actual control vector denoted by variables $u_x$ and $u_y$. The parameter $g$ is a dissipative constant (0.1 Nsm⁻¹), *Equations 3, 4* capture the first-order muscle dynamics with a time constant set to 0.1 s, and $f_{LOAD}$ represents the external load applied to perturb the point-mass (if any).

Defining the state vector as $z = [x, y, \dot{x}, \dot{y}, f_x, f_y]$, and the control vector $\boldsymbol{u} = [u_x, u_y]$, we have $\dot{z} = Az + Bu$, with:

$$A = \begin{bmatrix} 0 & 0 & 1 & 0 & 0 & 0 \\ 0 & 0 & 0 & 1 & 0 & 0 \\ 0 & 0 & -g/m & 0 & 1/m & 0 \\ 0 & 0 & 0 & -g/m & 0 & 1/m \\ 0 & 0 & 0 & 0 & -1/\tau & 0 \\ 0 & 0 & 0 & 0 & 0 & -1/\tau \end{bmatrix} \tag{5}$$

and

$$B = \begin{bmatrix} 0 & 0 \\ 0 & 0 \\ 0 & 0 \\ 0 & 0 \\ 1/\tau & 0 \\ 0 & 1/\tau \end{bmatrix} \tag{6}$$

The system was discretized by using a first-order Taylor series expansion over one time-step of $\delta t$: $A_d = I + \delta t A$, and $B_d = \delta t B$ ($I$ is the identity matrix). We used a discretization step of $\delta t = 0.01 s$.

Importantly, the state space representation of body dynamics was augmented with the x and y-coordinates of the two targets T1 and T2 (denoted by $x^*_{T1}, y^*_{T1}, x^*_{T2}, y^*_{T2}$), and the system was re-written in discrete-time equations as follows:

$$z_{(t+1)} = A_d z_{(t)} + B_d u_{(t)} + \varepsilon_t \tag{7}$$

The subscript $t$ is the time-step, and $\varepsilon_t$ is a Gaussian disturbance with zero-mean and covariance $\sum_\varepsilon = \sigma BB^T$, where $\sigma = 0.2$ is the scaling coefficient.

## Control policy

The control policy ($\Pi$) produces motor commands ($u_t$) as a function of the estimated state of the point-mass ($z_t$). The state-estimate is computed using Kalman filtering as a weighted sum of internal model prediction, and the sensory feedback represents the delayed state information ($z_{(t-\Delta)}$). The sensorimotor delays were set at $\Delta = 50 \, ms$ in the simulations. The details of the Kalman filtering can be found in the previous paper (*Crevecoeur et al., 2011*). Overall, motor commands are computed by a linear control policy:

$$u_t = \Pi(\hat{z}_t) = K_t \hat{z}_t \tag{8}$$

The control gains ($K_t$) determine how much each state variable can influence the motor command, and are computed to minimize a quadratic cost on the behavior ($J$), using standard dynamic programming procedures (Riccatti equations) described in previous literature (*Crevecoeur et al., 2020b*; *Crevecoeur et al., 2011*). So, the control policy can be formalized as finding the trajectory of the control gains '$K_t$' that minimize the expected costs on the entire sequence (i.e. a holistic plan):

$$\Pi^* = \underset{K_{t \in [0, N]}}{\operatorname{argmin}} J(\hat{z}, u) \tag{9}$$

Importantly, the control policy from *Equation 9*, receives information about both targets within a sequence as state feedback information, in addition to the continuous but delayed position, velocity, and forces on the end-effector.

## Cost function

As described above, the controller design minimizes a quadratic cost function that captures the intended behavior (in this case a two-reach sequence task). The cost function ($J$) was defined as a function of the state-estimate ($\hat{z}$), and the motor commands ($u$) used to control the point-mass. In a linear quadratic regulator (LQG), the control problem is to find a control sequence $\boldsymbol{u^*}$ which minimizes the expected value of the sum of J($\hat{z}$, u).

## Model with terminal costs

Typically, in the case of single reaches, the cost function penalizes the distance between the target and the actual position of the end-effector (terminal or trajectory errors), speed, and motor costs. But an efficient reaching movement in the context of a single reach does not ensure the efficiency of the sequence as a whole. Hence, we consider a composite cost function, that penalizes terminal errors between the end-effector and both targets, to optimize the sequence elements as a whole. Additionally, we can encourage swift or slower transfer between sequence elements by applying a cost on the velocity at the moment when the system approaches the first target (i.e. the intermediate goal). Let the duration of the sequence (N) be divided into two elements, where the first target (T1) should be reached at time-step 'N1', and the second target (T2) at time-step 'N2.' The holistic cost can be formulated as follows:

$$J\left(z, u\right) = J_{T1_{error}} + J_{T2_{error}} + J_{T1_{velocity}} + J_{T2_{velocity}} + J_{motorcost} \tag{10}$$

$$J_{T1_{error}} = w_1 \left[ \left(x_{(N1)} - x^*_{T1}\right)^2 + \left(y_{(N1)} - y^*_{T1}\right)^2 \right] \tag{11}$$

$$J_{T2_{error}} = w_1 \left[ \left(x_{(N2)} - x^*_{T2}\right)^2 + \left(y_{(N2)} - y^*_{T2}\right)^2 \right] \tag{12}$$

$$J_{T1_{velocity}} = w_2 \left[ \dot{x}^2_{(N1)} + \dot{y}^2_{(N1)} \right] \tag{13}$$

$$J_{T2_{velocity}} = w_3 \left[ \dot{x}^2_{(N2)} + \dot{y}^2_{(N2)} \right] \tag{14}$$

$$J_{motorcost} = w_4 \sum_{t=1}^{N2} \|u_t\|^2 \tag{15}$$

*Equations 11 and 12* penalize the terminal cost on the positional error relative to T1 and T2 at time-steps N1 and N2, respectively. An equal weight ($w_1 = 500$) was used for both terminal error terms. *Equations 13 and 14* penalize the squared velocity at each target, and the last *Equation 15* penalizes high motor output. A rapid transfer between two reaches in a sequence can be simulated by removing the penalty on the velocity at the first target. Hence, in this case, $w_2 = 0.1$, whereas a slower sequence involves slowing down significantly at the first target, which corresponds to a parameter set to $w_2 = 1$ in the simulations. In all cases, we set $w_4 = 10^{-4}$ as the coefficient on the motor costs term ($\|u_t\|^2$). Qualitatively similar results can be obtained with a broad range of parameter values ($w_1$, $w_2$, $w_3$, $w_4$).

## Model with temporal buildup costs

The terminal costs, described above, are sufficient to produce accurate movements with low endpoint errors, while neglecting the trajectory errors before the endpoint. Another possibility is to penalize deviations in the trajectory from the desired endpoint targets throughout the movement duration. Notably, by using temporal buildup of costs, the endpoint errors incur a higher costs, while incurring lower but non-zero cost in the movement duration before the endpoints. Such formulation has been recently applied to emulate human reaching control under uncertain environmental dynamics, in the context of elementary reaching movements (*Crevecoeur et al., 2019*; *Crevecoeur et al., 2020b*). To implement the temporal buildup costs model, the position and velocity terms in *Equations 11–14* were modified as follows:

$$J_{T1_{error}} = \sum_{t=1}^{N1} w_1 \left(t/N1\right)^{b1} \left[ \left(x_{(t)} - x_{T1}^*\right)^2 + \left(y_{(t)} - y_{T1}^*\right)^2 \right] \tag{16}$$

$$J_{T2_{error}} = \sum_{t=N1}^{N2} w_1 \left(\frac{t - N1}{N2 - N1}\right)^{b2} \left[ \left(x_{(t)} - x_{T2}^*\right)^2 + \left(y_{(t)} - y_{T2}^*\right)^2 \right] \tag{17}$$

$$J_{T1_{velocity}} = \sum_{t=1}^{N1} w_2 \left(t/N1\right)^{b1} \left[ \dot{x}_{(t)}^2 + \dot{y}_{(t)}^2 \right] \tag{18}$$

$$J_{T2_{velocity}} = \sum_{t=N1}^{N2} w_2 \left(\frac{t - N1}{N2 - N1}\right)^{b2} \left[ \dot{x}_{(t)}^2 + \dot{y}_{(t)}^2 \right] \tag{19}$$

Where the terms $(t/N1)^{b1}$ and $\left(\frac{t-N1}{N2-N1}\right)^{b2}$ assign a higher cost for the errors that occur when t=N1 and t=N2, respectively, otherwise allowing for non-zero but time-increasing costs. The parameters $b1 = 20$ and $b2 = 4$ determine the steepness of increase in the cost for a given state from starting time to the terminal times $N1$ and $N2$, respectively. Intuitively, these temporal buildup costs can encourage relatively straight point-to-point reaching as the deviations from the targets (T1 and T2) are compensated throughout the movement duration.

## Acknowledgements

We thank our lab members and Kevin cross for their feedback on different drafts. HTK was supported by the Fonds de la Recherche Scientifique (FRS-FNRS) Chargé de recherche Grant CR 252 (FC 043127). FC was supported by the FRS-FNRS Grant 1 .C.033.18 (FC 036239). This work was additionally supported by a Concerted Research Action of Université Catholique de Louvain (ARC; 'coAction').

## Additional information

### Funding

| Funder | Grant reference number | Author |
|---|---|---|
| Fonds De La Recherche Scientifique - FNRS | FC 043127 | Hari Teja Kalidindi |
| Fonds De La Recherche Scientifique - FNRS | FC 036239 | Frederic Crevecoeur |

The funders had no role in study design, data collection and interpretation, or the decision to submit the work for publication.

### Author contributions

Hari Teja Kalidindi, Conceptualization, Data curation, Formal analysis, Funding acquisition, Validation, Investigation, Visualization, Methodology, Writing - original draft, Writing - review and editing; Frederic Crevecoeur, Conceptualization, Resources, Supervision, Funding acquisition, Validation, Investigation, Methodology, Writing - original draft, Project administration, Writing - review and editing

**Author ORCIDs**
Hari Teja Kalidindi ⬤ https://orcid.org/0000-0003-2634-7953
Frederic Crevecoeur ⬤ https://orcid.org/0000-0002-1147-1153

## Ethics

All participants were neurologically healthy and gave informed consent according to a protocol approved by the ethics board at UCLouvain (Belgian registration number: B403201940375, and the ethics approval number: 2019/02MAY/196). Experiments did not exceed 2 h, and participants were compensated for their time.

Reviewer #1 (Public Review): https://doi.org/10.7554/eLife.96854.3.sa1
Reviewer #2 (Public Review): https://doi.org/10.7554/eLife.96854.3.sa2
Author response https://doi.org/10.7554/eLife.96854.3.sa3

# Additional files

## Supplementary files
• MDAR checklist

## Data availability

The data from simulations and experiments is available in the public repository 'figshare'.

The following dataset was generated:

| Author(s) | Year | Dataset title | Dataset URL | Database and Identifier |
|---|---|---|---|---|
| Kalidindi H, Crevecoeur F | 2024 | Task dependent coarticulation of motor sequences | https://doi.org/10.6084/m9.figshare.24282418.v1 | figshare, 10.6084/m9.figshare.24282418 |

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
