## [Editor Report · eLife assessment]

This **valuable** paper presents **convincing** evidence that changing the constraint of how long to stop at an intermediate target significantly influences the degree of coarticulation of two sequential reaching movements, as well as their response to mechanical perturbations. Using an optimal-control framework, the authors offer a normative explanation of how both co-articulated and separated sequential movement can be understood as an optimal solution to the task requirements.

---

## [Referee Report · Reviewer #1 (Public Review)]

In their paper, Kalidini et al. investigate why the motor system sometimes coarticulates movements within a sequence. They begin by examining this phenomenon in an optimal feedback controller (OFC) that performs reaching movements to two targets (T1 and T2). They show that coarticulation occurs only when the controller is not required to slow down at T1. When the controller must decelerate at T1, coarticulation does not occur. This observation holds true even though the controller has information about both targets in both scenarios. They test the same experiment on human participants and show that humans also coarticulate the reaches only when they are instructed to treat the first target as a via point. Both in human participants and OFC simulations, whenever the coarticulation is present, the long-latency response to perturbations during the first reach is also informed by the second target- suggesting that the information about the second target is already present in the circuitry that control the long-latency reflex.

All experiments and analyses are standard and clearly explained. Their analysis of long-latency as a measure of coarticulation of sequence items is highly interesting and broadly useful for future experiment design. They successfully demonstrate that one reason the motor system sometimes coarticulates movements is due to high-level instructions on how to execute the sequence. These high-level instructions can, in turn, determine how and to what extent information about future sequence items is utilized by the low-level controller that governs muscle activity. However, the precise interaction between high-level task demands and low-level controllers at the neural tissue level remains an open question.

---

## [Referee Report · Reviewer #2 (Public Review)]

Summary:

In this manuscript the authors examine the question of whether discrete action sequences and coarticulated continuous sequential actions can be produced from the same controller, without having to derive separate control policies for each sequential movement. Using modeling and behavioral experiments, the authors demonstrate that this is indeed possible if the constraints of the policy are appropriately specified. These results are of interest to those interested in motor sequences, but it is unclear whether these findings can be interpreted to apply to the control of sequences more broadly (see weaknesses below).

Strengths:

The authors provide an interesting and novel extension of the stochastic optimal control model to demonstrate how different temporal constraints can lead to either individual or coarticulated movements. The authors use this model to make predictions about patterns of behavior (e.g., in response to perturbations), which they then demonstrate in human participants both by measuring movement kinematics as well as EMG. Together this work supports the authors' primary claims regarding how changes in task instructions (i.e., task constraints) can result in coarticulated or separated movement sequences and the extent to which the subsequent movement goal affects the planning and control of the previous movement.

Weaknesses:

Although this work is quite interesting, it remains unknown whether there is a fundamental distinction between a coarticulated sequence and a single movement passing through a via point (or equivalently, avoiding an obstacle). The notion of a coarticulated sequence brings with it the notion of sequential (sub)movements and temporal structure, whereas the latter can really be treated as more of a constraint on the production of a single continuous movement. The authors suggest that these are not truly different kinds of movements at the level of a control policy, but this remains to be tested experimentally.

It also remains unclear for the theory of optimal feedback control as a whole where and how the cost function and constraints are specified to guide the optimization process. That is, presumably there is the ability for higher-level or explicit description of these constraints, but how they then become incorporated into a control policy remains unclear. With regard to the kind of multi-target constraints proposed here, in typical sequence tasks, while some movements become coarticulated, people also tend to form chunks with distinct chunk boundaries. This presumably means that there is at least some specification of the sequential ordering of these chunks that must exist beyond the control policy and that multiple control policies may still be warranted to execute an entire sequence (otherwise the authors' model might suggest that people can coarticulate forever without needing to exhibit any chunk boundaries). Hence, while the authors fairly convincingly show that a single control policy can lead to separated or coarticulated movements given an appropriate set of constraints, their work does not speak to where or how those constraints are specified, nor to how longer sequences are controlled.

---

## [Author Response]

The following is the authors’ response to the original reviews.

**Public Reviews:**

**Reviewer #1 (Public Review):**
Summary:In this paper, Kalidindi and Crevecoeur ask why sequential movements are sometimes coarticulated. To answer this question, first, they modified a standard optimal controller to perform consecutive reaches to two targets (T1 and T2). They investigated the optimal solution with and without a constraint on the endpoint's velocity in the via target (T1). They observed that the controller coarticulates the movements only when there is no constraint on the speed at the via-point. They characterized coarticulation in two ways: First, T2 affected the curvature of the first reach in unperturbed reaches. Second, T2 affected corrective movements in response to a mechanical perturbation of the first reach.Parallel to the modeling work, they ran the same experiment on human participants. The participants were instructed to either consider T1 as via point (go task) or to slow down in T1 and then continue to T2 (stop task). Mirroring the simulation results, they observed coarticulation only in the go task. Interestingly, in the go task, when the initial reach was occasionally perturbed, the long-latency feedback responses differed for different T2 targets, suggesting that the information about the final target was already present in the motor circuits that mediate the long-latency response. In summary, they conclude that coarticulation in sequential tasks depends on instruction, and when coarticulation happens, the corrections in earlier segments of movement reflect the entirety of the coarticulated sequence.EvaluationAmong many strengths of this paper, most notably, the results and the experiment design are grounded in, and guided by the optimal control simulation. The methods and procedures are appropriate and standard. The results and methods are explained sufficiently and the paper is written clearly. The results on modulation of long-latency response based on future goals are interesting and of broad interest for future experiments on motor control in sequential movement. However, I find the authors' framing of these results, mostly in the introduction section, somewhat complicated.The current version of the introduction motivates the study by suggesting that "coarticulation and separation of sub-movement [in sequential movements] have been formulated as distinct hypotheses" and this apparent distinction, which led to contradictory results, can be resolved by Optimal Feedback Control (OFC) framework in which task-optimized control gains control coarticulation. This framing seems complicated for two main reasons. First, the authors use chunking and coarticulation interchangeably. However, as originally proposed by (Miller 1956), the chunking of the sequence items may fully occur at an abstract level like working memory, with no motoric coarticulation of sequence elements at the level of motor execution. In this scenario, sequence production will be faster due to the proactive preparation of sequence elements. This simple dissociation between chunking and coarticulation may already explain the apparent contradiction between the previous works mentioned in the introduction section. Second, the authors propose the OFC as a novel approach for studying neural correlates of sequence production. While I agree that OFC simulations can be highly insightful as a normative model for understanding the importance of sequence elements, it is unclear to me how OFCs can generate new hypotheses regarding the neural implementation of sequential movements. For instance, if the control gains are summarizing the instruction of the task and the relevance of future targets, it is unclear in which brain areas, or how these control gains are implemented. I believe the manuscript will benefit from making points more clear in the introduction and the discussion sections.

We agree that chunking may occur at different levels that do not necessarily involve motor coarticulation. We clarified that our contribution is towards answering why sequence movements sometimes coarticulate, and how the way sequences are executed influences the representation of future goals in the sensorimotor system.

To address this point, we made the following modifications in the introduction:

Line 44:

“It remains unclear how future goals are integrated in the sensorimotor system. For rapid execution of a sequence, one possible solution is to represent multiple goals within low-level control circuits (3, 16), enabling the execution of several elements as a single entity, called “motor chunk”. Note that chunking can also occur at a higher level such as in working memory-guided sequences, which in this case may or may not involve the production of a movement (17, 18).”

Lines 50:

“Recent neural recordings in the primary motor cortex (M1) have shown no specific influence of future goals on the population responses governing ongoing action (19, 20). Specifically, Zimnik and Churchland (20) observed in a two-reach sequence task that, there was no coarticulation in sub-movement kinematics although the execution got faster with practice. Notably, M1 displayed separate phases of execution related activity for each sub-movement. Using a neural network model, they interpreted that sequence goals could be separated and serially specified to the controller from regions upstream of M1 (Figure 1A). These findings contrast with earlier studies showing coarticulation of sub-movements and whole sequence representations in M1 (21–23). As a result, it has been suggested that coarticulation and separation in rapid sequences may involve distinct computations: coarticulation possibly involves replacing sub-movements with a motor chunk, while separation possibly indicates independent control of each sub-movement with chunking at a higher-level (4, 20). Thus, there are unresolved questions regarding why sequential movements sometimes coarticulate, and how the representation of future goals in the sensorimotor system influences the way sequences are executed.”

With respect to the second part of your concern about OFC, we agree that this framework does not make direct prediction about the neural implementation and our statements required clarifications. The first link between the model and prediction about neural data follows from the observation that long-latency circuits participate in task-dependent sequence production, thus indicating that transcortical pathways must express this task dependency. The second link between our work and neural activities is by providing a counter argument to previous interpretation: indeed, Zimnik and Churchland argued that independent or “holistic” sequence production should be associated with different representations in monkey’s brain. In contrast we suggest that the same controller can flexibly generate both kinds of sequences, without implying a different structure in the controller, only a different cost-function. We thus refine the expectation about neural correlates of sequence representations by showing that it potentially relates to the encoding of task constraints.

To address this point, we added the following changes in the introduction and discussion:

Line 69 in Introduction:

“The theory of optimal feedback control (OFC) has been particularly useful in predicting the influence of numerous task parameters on the controller (27–34), thus reproducing goal-directed motor commands during both unperturbed movements and feedback responses to disturbances (30). OFC has been used in numerous studies to interpret flexible feedback responses occurring in the long-latency response period (30, 35).”

Line 454 in Discussion:

“Although OFC has been predominantly used as a behavioral level framework agnostic to neural activity patterns, it can shed light on the planning, state estimation and execution related computations in the transcortical feedback pathway (Takei et al.,). Using OFC, our study proposes a novel and precise definition of the difference to expect in neural activities in order to identify coarticulated versus independent sequence representations from a computational point of view. Because each condition (i.e., overlapping versus non-overlapping controllers as in Figure 2) was associated with different cost-functions and time-varying control gains, it is the process of deriving these control gains, using the internal representation of the task structure, that may differ across coarticulated and separated sequence conditions. To our knowledge, how and where this operation is performed is unknown. A corollary of this definition is that the preparatory activity (20, 50) may not discern independently planned or coarticulated sequences because these situations imply different control policies (and cost functions), as opposed to different initial states. Moreover, the nature of the sequence representation is potentially not dissociable from its execution for the same reason.”

**Reviewer #2 (Public Review):**
Summary:In this manuscript, the authors examine the question of whether discrete action sequences and coarticulated continuous sequential actions can be produced from the same controller, without having to derive separate control policies for each sequential movement. Using modeling and behavioral experiments, the authors demonstrate that this is indeed possible if the constraints of the policy are appropriately specified. These results are of interest to those interested in motor sequences, but it is unclear whether these findings can be interpreted to apply to the control of sequences more broadly (see weaknesses below).Strengths:The authors provide an interesting and novel extension of the stochastic optimal control model to demonstrate how different temporal constraints can lead to either individual or coarticulated movements. The authors use this model to make predictions about patterns of behavior (e.g., in response to perturbations), which they then demonstrate in human participants both by measuring movement kinematics as well as EMG. Together this work supports the authors' primary claims regarding how changes in task instructions (i.e., task constraints) can result in coarticulated or separated movement sequences and the extent to which the subsequent movement goal affects the planning and control of the previous movement.Weaknesses:I reviewed a prior version of this manuscript, and appreciate the authors addressing many of my previous comments. However, there are some concerns, particularly with regard to how the authors interpret their findings.

We thank the reviewer for their continued assessment of our work and for helping us to improve the paper. We are convinced that this and the previous review helped us clarifying our work considerably.

(1) It would be helpful for the authors to discuss whether they think there is a fundamental distinction between a coarticulated sequence and a single movement passing through a via point (or equivalently, avoiding an obstacle). The notion of a coarticulated sequence brings with it the notion of sequential (sub)movements and temporal structure, whereas the latter can be treated as more of a constraint on the production of a single continuous movement. If I am interpreting the authors' findings correctly it seems they are suggesting that these are not truly different kinds of movements at the level of a control policy, but it would be helpful for the authors to clarify this claim.

Indeed, this is our interpretation of the results/simulations. This suggestion can also be observed in Ramkumar et al., article on chunking. To clarify this, we added a statement in the discussion as follows:

Line 449:

“Notably, in the framework of optimal feedback control, an intermediate goal is equivalent to a via-point that constrains the execution of the sequence (similar to 13). It is thus possible that coarticulation in motor systems be processed similarly as other kinds of movement constraints, such as via-points, avoiding obstacles, or changes in control policies.”

(2) The authors' model clearly shows that each subsequent target only influences the movement of one target back, but not earlier ones (page 7 lines 199-204). This stands in contrast to the paper they cite from Kashefi 2023, in which those authors clearly show that people account for at least 2 targets in the future when planning/executing the current movement. It would be useful to know whether this distinction arises because of a difference in experimental methodology, or because the model is not capturing something about human behavior.

Thank you for raising this point. There are some differences between the study of Kashefi and colleagues (2023), and ours. Both studies looked into planning of more than one reach. In the study of Kashefi et al., the results of Figure 6 showed that in H2 condition, there was no significant curvature, and the curvature increases in H3 and H4 conditions (only in the 75ms dwell-time scenario). Note that H2 condition in their work meant the presentation of +2 target after the initiation of +1 reach. Hence, we think the GO task in our case should be compared to the H3 condition, resulting in similar curvature as in our study. These authors also showed that curvature increased even in the H4 condition (75 ms dwell). OFC also accommodates this observation, if we consider the relationship between the cost of intermediate goals and spatial location of the targets (see figure below, also added to Supplementary Figure 4). To see this, we performed additional 3 target simulations where the constraint on intermediate goal velocity (at T1 and T2) was varied to achieve similar dwell velocity at the intermediate targets (Supplementary Figure 4C). In this case, the hand curvature of the first reach differed while the dwell velocity was similar across T3 up and T3 down conditions, as may be instructed experimentally. Again, the task instructions and the spatial location of the future goals together determine how much the first reach components are influenced by the next ones, and this may impact several reaches ahead.

We added the following clarification in the result to describe this.

Line 199:

“It is worth noting that the OFC model can be generalized to longer sequences (10) through the incorporation of additional cost terms (in Equation 10 of Methods) and targets, enabling simultaneous planning for more than two targets. Simulations of a sample three-reach sequence (Supplementary Figure S4) revealed that, varying the cost of dwell velocity at intermediate targets (w2 and w3 parameters in Methods) caused a variation in control gains. Different amount of change in control gains can be expected for intermediate versus late targets (Supplementary Figure 4A). Notably, even when we used the same dwell velocity cost (w2 = w3 = 0), the observed velocity profiles were different between the two sequences towards different final targets (T3 up and T3 down) (Supplementary Figure 4B). We tested a condition in which both sequence reaches were forced to have similar dwell velocity profiles by increasing the dwell velocity costs in the sequence towards one of the targets (T3 down), while leaving this parameter unchanged for the other target (T3 up). In this scenario, T3 up sequence had the parameters (w2, w3) = (0, 0), while T3 down sequence had the parameters (0.8, 0.8). In this case, the curvature of the first reach was different, and predominantly occurred due to differences in K2 between the two sequence reaches (Supplementary Figure S4C). These simulations highlight that, planning for a longer horizon sequence can indirectly influence the curvature of early reaches, due to the interaction between intermediate dwell constraints, spatial arrangement of targets, and sequence horizon in a task dependent manner.”

(3) In my prior review I raised a concern that the authors seem to be claiming that because they can use a single control policy for both coarticulated and separated movement sequences, there need not be any higher-level or explicit specification of whether the movements are sequential. While much of that language has been removed, it still appears in a few places (e.g., p. 13, lines 403-404). As previously noted, the authors' control policy can generate both types of movements as long as the proper constraints are provided to the model. However, these constraints must be specified somewhere (potentially explicitly, as the authors do by providing them as task instructions). Moreover, in typical sequence tasks, although some movements become coarticulated, people also tend to form chunks with distinct chunk boundaries, which presumably means that there is at least some specification of the sequential ordering of these chunks that must exist (otherwise the authors' model might suggest that people can coarticulate forever without needing to exhibit any chunk boundaries). Hence the authors should limit themselves to the narrow claim that a single control policy can lead to separated or coarticulated movements given an appropriate set of constraints, but acknowledge that their work cannot speak to where or how those constraints are specified in humans (i.e., that there could still be an explicit sequence representation guiding coarticulation).

We thank the reviewer for raising this point. We do not dispute the statement that the controller needs to be set dependent on the constraints of the task that must be specified somewhere. In our view, this problem is similar to the question of how a cost-function (or a task representation) is transformed into a control policy in the brain, which is unknown in general. In the earlier version, our intention was to stress that separation can occur without necessarily implying that the goals be processed independently (as in Figure 1A and Zimnik 2021). To avoid confusion on this point, we modified this statement in the new version as follows:

Line 405:

“A straightforward interpretation could be that the stopping at the first target invoked a completely different strategy in which the control of the two reaches was performed independently (Figure 1A), effectively separating the two movements, whereas executing them rapidly could produce the merging of the two sub-movements into a coarticulated sequence. While this is conceptually valid, it is not necessary and the model provides a more nuanced view: both apparent separation or coarticulation of the two motor patterns can be explained within the same framework of flexible feedback control. These different modes of sequence execution still require proper specification of the task constraints in the model, such as number of intermediate steps, dwell-time, or velocity limit. Such specifications must be considered as input to the controller.”

**Recommendations for the authors:**

**Reviewer #1 (Recommendations For The Authors):**
Line 57: Distinct hypotheses.Line 209, The term "planned holistically" is confusing here. Seems like the authors suggest that the sequence is "planned holistically" as long as all sequence elements are given during the optimization process.

We changed the sentence as follows.

Line 218:“Overall, the model predicted that even if a feedback control policy was computed by optimizing the whole sequence over a long time-horizon, the requirements associated with intermediate goals determine how early in the sequence the second (future) target can influence the feedback controller”Line 336, It was not clear to me why the authors explained "the weak significant" results of PEC shortening in R0 given the nonsignificant values in R1.

We wanted to be transparent about whether changing the statistical analysis will lead to different interpretations, such as the sequence encoding even before long latency epochs. But we realized that it could lead to confusion and we deleted this sentence in the updated manuscript.

**Reviewer #2 (Recommendations For The Authors):**
About Weakness #2, to clarify this point the authors should either model and discuss what it would take for their model to account for multiple targets ahead, or else run a study to show that in this task people indeed only ever plan 1 target ahead.

Please see our response above (in Weakness #2).

I am still puzzled by why people would resist the perturbation more when they eventually have to move in the direction of the perturbation (e.g., p 10 lines 313-314). Perhaps this is simply due to the geometry of the task, but it could also depend on what participants were trying to accomplish in the experiment. To help clarify this, the authors should report exactly what instructions were given to participants in each task condition.

The simulations suggest that the observed perturbation movements are an optimal way to perform the task given the task constraints on accuracy, control effort and constraints at intermediate goals. The intuition is that modulating the acceleration at the intermediate goal is preferred rather than missing it. This however depends on the cost parameter.

Below, in Author response figure 1, we show the simulations by varying the accuracy requirements at intermediate goal and the total motor cost parameters. Clearly, as expected, increasing the cost on accuracy of the intermediate reach, or decreasing the cost on motor output modulated the hand deviation (simulations not included in the article).

**Author response image 1. sa3fig1:** Impact of movement costs (motor effort and intermediate goal reach errors) on the hand path following a mechanical perturbation.

Our observation suggests that participants’ behaviour agreed with the interpretation that can result from the model. We clarified the exact instructions in the methods section. Note that the instructions were given at the beginning of the task and did not differ across the different conditions involving changes in the location of T2 or perturbation direction:

Line 594:

Participants were given the following instructions verbally: “Wait in the starting circle until you receive a GO signal, where the target circles turn red and you will simultaneously hear a beep sound. When the circles turn red, react quickly, move as soon, and as straight as possible to target 1 and then move to target 2. You will get two points at the end of the trial if you reach T1 in the prescribed time window and then move to T2, and in all other cases you will not receive any points. Importantly, once you reach T1 you should try to come out of it quickly. If you stay in T1 for more than 150 ms then T2 will disappear and you will receive only one point. Additionally, in some trials, a force will perturb your hand towards the right or left direction randomly while moving towards T1. The instructions remain the same in the presence of perturbations. Try to score as many points as you can.”

Additionally, we added the following lines in the results description:

Line 284:

“The influence of second target on the lateral hand deviation was qualitatively similar to that observed in model simulations, and counterintuitive to what we might expect without the help of the model simulations. As observed in the model simulations (see also Supplementary Figure S2), lateral hand deviation was smaller when the perturbation was in the direction of the second target (T2) and vice-versa. This was consistent for both rightward and leftward perturbation conditions. Both the model and humans expressed this strategy that can be seen as an emergent feature of efficient feedback control during production of movement sequences. Additionally, even though behavior was reproduced in simulations, changing the cost on control effort and/or accuracy of intermediate reaches could modulate the sequencedependent changes in curvature.”

I am not sure if "the data and code for simulations can be provided by the corresponding author" satisfies the eLife/PLoS software guidelines (i.e., that it be deposited in a public repository).

Thank you for pointing this out. This sentence was added by mistake.

We modified this statement in the updated manuscript.

“The data and code from simulations and experiments is available in the public repository ‘figshare’ in the following link: https://doi.org/10.6084/m9.figshare.24282418.v1.